# Therapeutic Potential of Natural Products in Treatment of Cervical Cancer: A Review

**DOI:** 10.3390/nu13010154

**Published:** 2021-01-05

**Authors:** Seung-Hyeok Park, Minsun Kim, Somi Lee, Woojin Jung, Bonglee Kim

**Affiliations:** 1Department of Korean Obstetrics & Gynecology, Kyung Hee University Korean Medicine Hospital, Seoul 02447, Korea; baomose@khu.ac.kr; 2College of Korean Medicine, Kyung Hee University, Seoul 02447, Korea; red_pomme@khu.ac.kr (M.K.); mayleee0110@khu.ac.kr (S.L.); wj0119@khu.ac.kr (W.J.); 3Korean Medicine-Based Drug Repositioning Cancer Research Center, College of Korean Medicine, Kyung Hee University, Seoul 02447, Korea

**Keywords:** cervical cancer, dietary natural products, apoptosis, angiogenesis, metastasis, resistance, microRNA

## Abstract

Cervical cancer is the fourth most common cancer among women worldwide. Though several natural products have been reported regarding their efficacies against cervical cancer, there has been no review article that categorized them according to their anti-cancer mechanisms. In this study, anti-cancerous natural products against cervical cancer were collected using Pubmed (including Medline) and google scholar, published within three years. Their mechanisms were categorized as induction of apoptosis, inhibition of angiogenesis, inhibition of metastasis, reduction of resistance, and regulation of miRNAs. A total of 64 natural products suppressed cervical cancer. Among them, *Penicillium sclerotiorum* extracts from *Cassia fistula* L., ethanol extracts from *Bauhinia variegate candida*, thymoquinone obtained from *Nigella sativa*, lipid-soluble extracts of *Pinellia pedatisecta* Schott., and 1′S-1′-acetoxychavicol extracted from *Alpinia conchigera* have been shown to have multi-effects against cervical cancer. In conclusion, natural products could be attractive candidates for novel anti-cancer drugs.

## 1. Introduction

Cervical cancer, which is a cancer arising from the cervix, is characterized by abnormal vaginal bleeding, vaginal discharge, pelvic pain, or pain during sexual intercourse [1]. Currently, cervical cancer is the fourth most common cancer among women in the world [2]. According to Globocan 2018, the prevalence rate of cervical cancer is 3.2% of all cancers. The main treatments for cervical cancer are surgery such as pelvic lymphadenectomy and radical hysterectomy, radiotherapy, and chemotherapy [3]. Another therapy is targeted therapy, which regulates epidermal growth factor receptor (EGFR) [4,5] and cyclooxygenase-2 (COX-2) [6,7] for treating cervical carcinoma. However, these treatments showed possible side effects and complications: Surgery could cause bleeding, damage to the organs around the surgery, and a risk of clots in the deep veins of the legs, radiotherapy could yield menopause, infertility, discomfort, or pain with intercourse, and the side effects of chemotherapy may affect not only cancer cells but also rapidly dividing cells in systems of the whole body [8,9]. Moreover, the drugs that are usually prescribed for cervical cancer showed several side effects and drug resistance [10]. Cisplatin, which is one of the most effective anticancer drugs, has resistance capacity by a self-defense mechanism [11]. 5-fluorouracil (5-FU) is also reported for resistance and side effects when it comes to cervical cancer patients [12]. Thus, we have focused on discovering a new potent treatment for cervical cancer from natural products.

Natural products extracted from living organisms including plants and animals have several active ingredients, which are reported to be attractive alternatives to chemotherapeutic drugs or suitable for combined use with chemotherapeutic drugs [13,14]. For example, purified flaxseed hydrolysate (PFH), extracted from Lignan, induces apoptosis and inhibits angiogenesis and metastasis on HeLa cells [15]. Thymoquinone from *Nigella sativa* also showed apoptotic effect and anti-proliferation in SiHa and CaSki cells. Such natural products include *Bauhinia variegate candida* ethanol extracts, Praeruptorin-B, and well-known tea.

MicroRNA (miRNA, miR) are involved in the pathological development and metastasis of cancer [16,17]. Several natural products showed an anti-cancer effect by regulation of cancer-related miRNAs. Our team reported that *Spatholobus suberectus* Dunn extract induces apoptosis by regulation of miR-657/activating transcription factor 2 (ATF2) in U266, U937 cells [18]. Another natural product, *Salvia miltiorrhiza*, showed an anti-cancer effect via regulation of miR-216b [19]. 1′S-1′-acetoxychavicol acetate (ACA) from *Alpinia conchigera* has been reported to induce apoptosis on SiHa and CaSki cells by targeting SMAD4 and miR-210. Targeting miRNAs with natural products could be a promising strategy for cervical cancer [20]. However, there have been no studies that organize the mechanisms, efficacy, and concentration of natural products for cervical cancer in last five years. In this present study, we aim to review the nonclinical studies about the anti-cancer mechanisms of natural products. The natural products were organized by their mechanisms including apoptosis, anti-metastasis, anti-angiogenesis, resistance, and microRNA regulation.

## 2. Methods

We collected the relevant experimental literature published in recent three years elucidating the anticancer effects of natural products on cervical cancer using PUBMED (including Medline) and the Google Scholar database. Upon searching for the appropriate studies, we used ‘natural product, cervical cancer’ as keywords. After completing the initial search, we removed duplicates and non-English literature. Chemical structures of compounds derived from natural products were cited from the NCBI PubChem website.

## 3. Results

### 3.1. Apoptosis

Apoptosis is a unique form of cell death and is an important process that regulates the homeostasis of cell survival [21]. Apoptosis eliminates potentially cancerous cells and this process is caused by atrophy of cells, synthesis of new proteins, and cell suicide genes; also, it has a great influence on the malignant phenotype [22]. For this reason, apoptosis is used as an anticancer mechanism for cancer research. A total of 47 studies have been performed to elucidate apoptosis-mediated anti-cancer pathway of natural products in Hela and SiHa cells. A total of 54 natural products were reviewed.

#### 3.1.1. Compounds

Among the natural products, 35 compounds showed an apoptotic effect against cervical cancer (Table 1). The chemical structures of the compounds are shown in Figure 1.

Li et al. showed that 4w of DHA-Based acylhydrazones induces anti-cancer activity with an IC_50_ value of 2.21 µM for 48 h in Hela cells [23]. Interestingly, 4w was about 17-fold more active than that of the parent compounds. Vishnu et al. reported that anthocyanins, extracted from purple root tubers and leaves of *Ipomoea batatas*, was shown to promote CFP/YFP activity in HeLa cells [24]. The treatment was done at concentrations of 100 and 200 µg/mL for 48 h. The result showed the induction of apoptosis, cell cycle arrest, and alteration of cell morphology by the treatment. Moreover, the leaf anthocyanins exhibited significantly higher activity against cervical cancer cells. Arborinine isolated from *Glycosmis parva* upregulated caspase-3, -7, and downregulated Bcl2-L1 at a dose of 110 μg/mL for an incubation time of 24 h in HeLa cells [25]. This result suggested that induction of apoptosis and inhibition of migration were involved in anti-cancer effect of arborinine. Arborinine also inhibited tumor spheroid growth than chemotherapeutic drugs such as bleomycin, gemcitabine, and cisplatin. Expression of p15, p53, and Bax were increased and cyclin D1, Bcl-2, MMP-2, -9, β-catenin, TCF7, and c-Myc were decreased by treatment of β-elemene extracted from *Curcuma zedoaria* in SiHa cells at dose of 30 μg/mL, 40 μg/mL, and 50 μg/mL for 24 h, 48 h, and 72 h [26]. This result suggested that induction of cell cycle arrest and inhibition of migration are involved in the pathway. In addition, β-elemene inhibited cell proliferation and invasion and induced apoptosis via attenuation of the Wnt/β-catenin signaling pathway in cervical cancer cells. Copper oxide nanoparticles synthesized by *Azadirachta indica*, *Hibiscus rosasinensis*, *Murraya koenigii*, *Moringa oleifera*, and *Tamarindus indica* induced apoptosis and inhibited oxidant in HeLa cells at doses of 2 μg/mL, 5 μg/mL, 10 μg/mL, 25 μg/mL, 50 μg/mL, and 100 μg/mL for 48 h [27]. Expression of p53, cytochrome c, caspase-3, -7, -9, and PARP were increased and Bcl-2 and NF-κB were decreased in HeLa cells by treatment of curcumin-loaded TPGS/F127/P123 mixed polymeric micelles isolated from *Curcuma longa* at dose of 2 μg/mL for 48 h [28]. Through the mechanisms, this induced apoptosis, mitochondria-mediated apoptosis, and cell cycle arrest. In vivo, mice were provided 25 mg/kg 11 times in 2-day intervals and in the curcumin-loaded TPGS/F127/P123 group, this could efficiently inhibit the growth of cervical cancer cells. Emodin reduced activation of HOCl/OCl- and p-Akt and inhibited NO- and O2- at concentrations of 46.3 μg/mL, 92.8 μg/mL, and 185 μg/mL for 6 h, 12 h, and 24 h for the treatment of SiHa and C33A cells [29]. The result showed that emodin was related to induction of cell cytotoxicity, apoptosis, oxidative stress, and DNA damage. When HeLa cells were treated by epifriedelinol isolated from *Aster tataricus* and *Vitex peduncularis*, the level of caspase-3, -8, and -9 were increased and Bcl-2, -xL, survivin, and actin were decreased at doses of 50 μg/mL, 100 μg/mL, 250 μg/mL, 500 μg/mL, and 1000 μg/mL for 72 h [30]. Through the mechanisms, it reduced cell viability and induced apoptosis by downregulating the expression of anti-apoptotic protein and enhancing the expressions of pro-apoptotic protein. In addition, it altered the ratio of pro-apoptotic to anti-apoptotic proteins. Eugenol from *Syzygium aromaticum* was treated to HeLa cells at doses of 12.5 µM and 25 µM for 24 and 48 h [31]. The treatment upregulated Bax, PARP, caspase-3, and ROS and downregulated both Bcl-2 and XIAP in HeLa and SiHa cells. The result showed the inhibitory efficacy of eugenol by changing the cell viability in a time- and dose-dependent manner with consistent morphological changes. In the study of Chen et al., the treatment of icaritin, which derived from *Epimedium*, increased ROS, Bax, c-caspase-3, and -9 and decreased Bcl-2 and XIAP in both HeLa and SiHa cells [32]. In the treatment, HeLa cells were treated at concentrations of 12.5 µM and 25 µM for 24 h, 48 h, and 72 h, and SiHa cells were treated at 12.5 µM and 25 µM for 24 h, 48 h, and 72 h. This result indicated that icaritin promoted cancer cell death via induction of extensive oxidative DNA damage, which subsequently resulted in large numbers of DNA strand breaks and the activation of the intrinsic apoptotic pathway. Kuo et al. reported the efficacy of juncusol, which originates from *Juncus inflexu*, against HeLa, SiHa, and CaSki cells [33]. With concentrations of 1 µM, 3 µM, 10 µM, and 30 µM for 24 h, 48 h, and 72 h, the treatment induced cell apoptosis and inhibited cell proliferation. Cell cycle analysis showed an increase in G2/M and sub G1 cell populations after juncusol treatment. Furthermore, caspase-3, -8, and -9 were observed to be activated in HeLa cells, suggesting that the treatment induced apoptosis. Moreover, juncusol inhibited tubulin polymerization, as well as EGFR activation, this might suggest that juncusol induced a G2/M-phase cell cycle arrest and inhibited cell migration. Ma et al. evaluated the anti-cancer activity of methyl protodioscin, isolated from *Polygonatum sibiricum*, which led to cell apoptosis in HeLa cells by increasing G2/M phase and ROS [34]. Methyl protodioscin was used at concentrations of 18.31 µM, 40 µM, and 49 µM for 24 h. This treatment had efficacy on inducing alteration of cell morphology and cell cycle arrest as well as inhibiting cell proliferation. Al-Otaibi et al. showed cytotoxicity and apoptosis enhancement in Hela cells by co-administration of mitomycin C (MMC) in ginger (Gi) and frankincense (Fr) oil [35]. When treated at a dose of 10 µg/mL for 24 h, Fr-MMC endured the nuclear apoptosis in HeLa cells at a lower concentration compared to Gi-MMC. Moreover, mixing MMC with Gi-NE and Fr-NE considerably improved its cytotoxicity on HeLa cells. Latif et al. observed antitumor activity of naringenin oxime and oxime ether derivatives [36]. By the compounds, HeLa cells were treated at concentration of 12 µM and 24 µM for 24 h, and SiHa cells were treated at doses of 18 µM, and 36 µM for 24 h. As a result, compound 6 was shown to have an apoptotic effect on HeLa and SiHa cells by activating caspase-3. Moreover, cell cycle analysis suggested that compound 6 caused an increase in the subG1 phase and induced apoptosis in HeLa and SiHa cells. Induction of caspase-3 and -7 and inhibition of aldolase A, alpha-enolase, pyruvate kinase, and glyceraldehyde 3-p-dehydrogenase in both HPV16 and SiHa cells were observed after the exposure of nitensidine B isolated from leaves of *Pterogyne nitens* Tul [37]. Cells were treated at doses of 30 µM, 60 µM, and 120 µM for 6 h, 12 h, and 24 h. The result of study suggested that nitensidine B had efficacy on inducing apoptosis and inhibiting glycolysis. Notoginsenoside R7 isolated from *Panax notoginseng* upregulated Bax, p-PTEN, and Akt and downregulated Bcl-2, -XL, caspase-3, -9, and raptor in HeLa cells at the densities of 5 µM, 10 µM, 20 µM, and 40 μM for 24 h, 36 h, and 48 h [38]. In vivo, tumor weight was reduced by approximately 28% and 52%, respectively in 5 mg/kg/day and 10 mg/kg/day treated mice compared with a control group. Through the mechanisms, it induced apoptosis and inhibited proliferation. In conclusion, notoginsenoside R7 could be used for the treatment of cervical cancer and other PI3K/PTEN/Akt/mTOR signaling-associated tumors. The exposure of osthole derived from *Cnidiummonnieri* (L.) Cusson on HeLa, SiHa, C-33A, and CaSki cells upregulated the level of Bax, c-caspase-3, -9 proteins, E-cadherin, and H2AX [39]. However, osthole downregulated of Bcl-2, MMP-2, -9, β-catenin, vimentin, N-cadherin, IKKα, p-IKKα, p65, p-p65, p50, and NF-κB. This mechanism was made by osthole at concentrations of 40 µM, 50 µM, 80 µM, 100 µM, 120 µM, 150 µM, 160 µM, 200 µM, and 240 µM for 24 h and 48 h. The result showed that osthole had an efficacy of inducing apoptosis, as well as inhibiting cell proliferation, cell viability, cell migration, and cell invasion. Physcion reduced activation of HOCl/OCl- and p-Akt at 43.8 μg/mL, 87.5 μg/mL, and 175 μg/mL for 6 h, 12 h, and 24 h for the treatment of SiHa and C33A cells [29]. The result showed that physcion was related to the induction of cell cytotoxicity, apoptosis, oxidative stress, and DNA damage. Phyto-synthesis of silver nanoparticles (AgNPs) isolated from garlic, green tea, and turmeric reduced cell viability and induced apoptosis and inhibited oxidant in HeLa cells [40]. Expression of free radicals was decreased through the treatment of AgNPs at doses of 2 μg/mL, 5 μg/mL, 10 μg/mL, 25 μg/mL, 50 μg/mL, and 100 μg/mL for 48 h. The AgNPs from turmeric extracts were the most excellent antioxidant and had great cytotoxicity activity compared to that synthesized using other extracts. Combination of paclitaxel and piperine, extracted from the plant *Piper nigrum*, was shown to promote the level of Bax, Bcl-2, c-PARP, and caspase-3 [41]. Additionally, the treatment decreased p-Akt and Mcl-1 in both HeLa and PTX cell. Cells were treated at a concentration of 50 µM for 6 h, 24 h, and 72 h. The result suggested induction of apoptosis was involved in the pathway. Zhang et al. reported that Hela cells showed increased EGFP, ROS, Bcl-2, cytochrome c, Apaf-1, caspase-3, and -9 and decreased c-Myc and hTERT after treated by prenylflavonoid C1 and C5, which derived from *Mallotus conspurcatus* [42]. These mechanisms were made at a concentration of 30 μM, for 24 h. The result indicated that the treatment was associated with induction of mitochondrial dysfunction, cytotoxicity, and apoptosis as well as inhibition of telomerase activity. Expression of JNK, p38, PERK, ATF4, Bax, caspase-3, -8, -9, and PARP were increased and Bcl-2 was decreased by treatment of protodioscin isolated from *Dioscoreae rhizome* in HeLa cells and C33A at the density of 4 μM for 24 h and 48 h [43]. Moreover, protodioscin was proved to induce ROS and the ER stress pathway. Consequently, protodioscin induced ER stress-dependent apoptosis, which could be mediated by the activation of the JNK and p38 pathways and mitochondrial dysfunction in the pathway. Chatterjee et al. showed the efficacy of two polyphenols, resveratrol and pterostilbene, against human HeLa cells [44]. Expression of activated caspase-3 was upregulated and PCNA and VEGF were downregulated in both PC1 and HPV E6 cells, at a concentration of 30 µM for 48 h. Further in vivo studies on TC1 tumors developing in mice indicated that treatment with either resveratrol or pterostilbene could significantly inhibit tumor development, with 1 mM for five days. This result suggested that resveratrol and pterostilbene induced cell cycle arrest and inhibited tumor growth in cervical cancer cell lines. Chen et al. reported that Tf-CT-ME, isolated from the plant *Tripterygium wilfordii*, upregulated c-caspase-3 and downregulated Bcl-2/Bax in HeLa cells [45]. The mechanisms were made by the treatment to HeLa cells, at doses of 0.5 µg/mL, 1 µg/mL, and 2 µg/mL for 24 h. This result meant that Tf-CT-ME induced cell cycle arrest and anti-proliferation. Thymoquinone isolated from *Nigella sativa* upregulated Bax and E-cadherin and downregulated Bcl-2, Twist1, and vimentin in SiHa and CaSki cells at doses of 10 μM, 20 μM, and 40 μM for 24 h, 36 h, and 48 h [46]. The expression of PARP, caspase-3, -9, Bax, and Bcl-2 in CaSki and SiHa cells were nearly unaffected by thymoquinone. It suggested thymoquinone effected on human cervical cells by other mechanisms. Consequently, inhibition of invasion and growth, probably via Twist1/E-cadherin/EMT or/and Zeb1/E-cadherin/EMT, and induction of apoptosis were involved in the process. Triphala was a combination of the fruity part of *Terminalia chebula* Retz., *Terminalia bellerica* Roxb., *Phyllanthus emblica* Linn., and it was shown to inhibit proliferation and induct apoptosis in HeLa cells [47]. The IC_50_ value against HeLa cells was 95.56 ± 8.94 μg/mL. This process was made by increasing ERK and p53 and decreasing c-Myc, cyclin D1, p-Akt, p-NF-κB, p56, p-p44/42, and MAPK at a dose of 25 μg/mL for 48 h. In conclusion, triphala had great promise for treating cervical cancers. The level of RSU1 and GAPDH were increased in CaSki and SiHa cells by treatment of 1′S’-1′-acetoxychavicol acetate (ACA) isolated from *Alpinia conchigera* at doses of 20 μM and 30 μM for 6 h, 12 h, and 48 h [48]. This result suggested that induction of apoptosis and reduction of cell viability were involved in the pathway. Downregulation of miR-629 enhanced sensitivity toward ACA and it indicated miR-629 could also play a role in regulating response toward anticancer agents. 2D and 3O of oleanolic acid and glycyrrhetinic acid derived from *Ligustri Lucidi Fructus* and *Glycyrrhiza uralensis* elevated the level of ROS in a concentration and time-dependent manner in HeLa cells [49]. 3O presented the strongest cytotoxicity on HeLa cells among the compounds. 2D was used at doses of 2 µM and 4 µM for 24 and 48 h, and 3O was used at 1 µM and 2 μM, for 48 h. Both compounds had an efficacy on inducing cell apoptosis, autophagy, alteration of cell morphology and cytotoxicity, and inhibiting proliferation. Hassan et al. reported that 4f, the compound of 3,5,4′-trimethoxystilbene and 5,6,7-trimethoxyflavone, activated cytotoxic and apoptotic activity in HeLa cells [50]. There were morphological changes of the nuclei in HeLa cells at a dose of 10 µM, after 48 h. In addition, time- and dose-dependently increased fluorescence signal meant that cell death was triggered in proportion. In conclusion, the study showed that the compound 4f triggered cell death through induction of apoptosis in HeLa cells. 5′-*epi*-SPA-6952A derived from *Streptomyces diastatochromogenes* upregulated the activation of Bax/Bcl-2, cytochrome c, caspase-3, -9, c-PARP, and p53, and downregulated MMP [51]. The mechanisms were observed at concentrations of 2 µg/mL, 4 µg/mL, 8 µg/mL, and 16 µg/mL, for 24 h. The treatment showed induction of apoptosis, cell cycle arrest, and alteration of cell morphology as well as inhibition of proliferation and migration.

Al-Otaibi et al. investigated the anticancer efficacy in HeLa cells upon co-administration of MMC and Gi and Fr oils [35]. The potential cytotoxicity of co-administrated formulas on HeLa cells were dose-dependently and remarkably greater than those of free MMC. Endurance of Fr-MMC to the nuclei apoptosis was at a lower concentration. Furthermore, Fr-MMC was reported to have the least % of florescence uptake compared to Gi-MMC. Chatterjee et al. observed the in vitro anti-cancer activity and in vivo anti-tumor potential of resveratrol and pterostilbene on HeLa cells [44]. The result of in vitro showed cytotoxic potential of both compounds, exhibiting the ability to inhibit viral oncogene E6. In vivo analysis showed inhibition of E6 and VEGF tumor protein levels. Moreover, the decline in tumor size in pterostilbene was related to activated caspase-3, while resveratrol-treated mice was related to arrest of cell cycle. The study of Chen et al. observed the anti-tumor efficacy of Tf-CT-MEs, by using transmission electron microscopy, dynamic light scattering, flow cytometry, and MTT assay [45]. Moreover, intratumor penetration was discovered by 3D tumor spheroid as the model. The result showed that Tf-CT-ME had higher apoptotic effects than those treated by CT-MEs.

Most studies on apoptosis of compounds were confirmed by mechanisms such as caspase, Bcl-2, Bax, ROS, PARP, and p-Akt. Among the mechanisms, the three most important were the regulation of caspase, Bcl-2, and Bax. Significant loss or inactivation of caspase impairs the induction of apoptosis, leading to a dramatic imbalance in growth dynamics, ultimately causing abnormal growth of human cancer [52]. Of the 30 compounds that induce apoptosis in cervical cancer, caspase-inducing compounds were the most common out of 19, and 16 of them induced caspase-3 expression. Caspase-9 was induced in eight compounds, caspase-7, -8, and c-caspase-3 were expressed in three, and c-caspase-9 in two. Bcl-2, which affects apoptosis and drug resistance [53], was regulated in 13 compounds, and Bax, a pro-apoptotic molecule that also functions as a tumor suppressor [54], was regulated in 10 compounds. There were nine compounds that inhibit Bcl-2 and two compounds that increase it, eight compounds that increase Bax, and two compounds have apoptosis by controlling the ratio of Bax and Bcl-2. Most of the studies were performed in vitro, but there were some studies performed in vivo [28,38,44]. A large number of studies have revealed the mechanism of apoptosis, but few studies have found it [23,27,35,50]. In addition, there have been studies where the concentration is too high, and toxicity is a concern [24,25,29,30,39,45].

#### 3.1.2. Extracts

The 19 natural extracts have been found to induce cell death in cervical cancer (Table 2). Swanepoel et al. showed that aqueous extracts of *Anemone nemorosa* caused a delay in the early mitosis phase of the cell cycle [55]. Apoptosis was confirmed through fluorescent staining with annexin V-FITC. The treatment with IC_50_ concentration of 20.33 ± 2.480 µg/mL elevated the expression level of PS translocation, c-caspase-3, -8, and ROS in 24 h. In 48 h, however, the level of MMP and ROS were suppressed. The result of this study indicated that induction of apoptosis and autophagy and inhibition of proliferation were induced in the pathway. Aril extracts isolated from *Strelitzia nicolai* induced apoptosis and inhibited oxidant in HeLa cells at a concentration of 250 μg/mL for an incubation time of 24 h, 48 h, and 72 h [56]. The aril extracts decreased cell viability by 52% and induced apoptosis in HeLa cells. Additionally, aril extracts produced a higher radical scavenging activity than the bilirubin standard. Ethanol extracts from *Astragalus membranaceus*, *Angelica gigas*, and *Trichosanthes kirilowii* upregulated c-caspase-3, -8, and PARP-1 and downregulated Bax, cyclin D, CDK2, CDK4, CDK6, and p27 in HeLa cells at doses of 100 μg/mL, 200 μg/mL, and 400 μg/mL for 24 h and 48 h [57]. This result suggested that induction of apoptosis, cell cycle arrest, and reduction of cell viability are involved in the pathway. However, ethanol extracts did not affect the intrinsic mitochondria-mediated apoptosis pathway in HeLa cells. Expression of c-caspase-3, -8, RIP, and TNF-R1 were increased and MMP-2 and-9 were decreased in HeLa cells by treatment of ethanol extracts from *Bauhinia variegate candida* at dose of 15 μg/mL for an incubation time of 24 h [58]. Controlling mechanisms, it could reduce cell viability and inhibit migration and induct apoptosis. In conclusion, it contained components with potential tumor-selective cytotoxic action. Ethanol extracts isolated from *Botryidiopsidaceae* species induced apoptosis and inhibited oxidant, proliferation, migration, and invasion in HeLa cells [59]. It upregulated p53 and c-caspase-3 and downregulated Bcl-2 at doses of 6.25 μg/mL, 12.5 μg/mL, 25 μg/mL, and 50 μg/mL for an incubation time of 24 h. In conclusion, inhibitory effects of ethanol extracts on migration and invasion might occur via the modulation of genes related to the processes of cellular invasion and migration. Ethanol extracts from *Chloromonas* species upregulated c-caspase-3 and p53 and downregulated Bcl-2 in HeLa cells at dose of 12.5 μg/mL and 25 μg/mL for an incubation time of 24 h and 72 h [60]. The result meant that ethanol extracts dealt with cancer cells by increasing the pro-apoptotic protein and reducing the anti-apoptotic protein. It suggested that induction of apoptosis through the modulation of apoptosis associated genes and inhibition of proliferation and oxidant were involved in the pathway. Ethanol extracts from *Dendrobium chrysanthum* upregulated Bax, and p53 and downregulated Bcl-2 in HeLa cells at the density of 450 μg/mL for an incubation time of 24 h [61]. This suggested that induction of apoptosis, DNA fragmentation, and ROS-altered cell morphology were involved in the pathway. In addition, the anticancer potential of the *Dendrobium chrysanthum* was mediated through p53-dependent apoptosis. In vivo, antitumor activity exhibited a significant increase in the life span of Dalton’s lymphoma-bearing mice with significant decrease in abdominal size along with reduced tumor ascites. Ethanol extracts from *Rhamnus sphaerosperma* var. *pubescens* (EERs) reduced activation of HOCl/OCl- and p-Akt at 25 μg/mL, 50 μg/mL, and 100 μg/mL for 6 h, 12 h, and 24 h for the treatment of SiHa and C33A cells [29]. The result showed that EERs were related to the induction of cell cytotoxicity, apoptosis, oxidative stress, and DNA damage. Ma et al. reported that EAEG, ethyl acetate extracts of *Gynura formosana* Kitam. leaves, exhibited antioxidant, anti-inflammatory, and autophagy-mediated inhibition of cell proliferation activity on HeLa cells [62]. The increased levels of LC3-II/LC3-I and decreased levels of P62/GAPDH and MCM7/GAPDH were observed at a concentration of 30 µg/mL for 72 h. Kuriakose et al. reported that ethyl acetate extracts of *Penicillium sclerotiorum*, isolated from *Cassia fistula* L., had cytotoxic activity [63]. At doses of 5 μg/mL, 25 μg/mL, and 50 μg/mL for 24 h, it arrested cells at S and G2/M phase of the cell cycle in a dose-dependent manner. Annexin V/propidium iodide double staining showed apoptosis more than necrosis. Moreover, the decreased Bcl-2 and increased Bax, p53, and Apaf-1 support apoptotic cell death. Ethyl acetate extracts from *Streptomyces* species upregulated caspase-3, -9, Bax, and LC3-II and downregulated PARP, LC3-I, Beclin1, and p62 in SiHa cells at concentrations of 20 μg/mL, 40 μg/mL, and 60 μg/mL for 24 h [64]. Half of the SiHa cells underwent death at a concentration of 20 μg/mL. This induced altered cell morphology and apoptosis, autophagy, and inhibited proliferation in the pathway. Consequently, ethyl acetate extracts-induced Bax-dependent mitochondrial permeabilization plays the role of an initiator of intrinsic pathway of apoptosis. Expression of p53 was increased and cyclin D, E, p21, and survivin were decreased in SiHa cells by treatment of blueberry extracts at the density of 50 mg/mL for an incubation time of 24 h with 4 Gy radiotherapy [65]. Through the mechanisms, it enhanced radiotherapy. Lipid-soluble extracts from *Pinellia pedatisecta* Schott. upregulated β-catenin, c-Myc, cyclin D1, PPAR1, and downregulated Th2 and Th17 in HPV and TC-1 cells at a dose of 500 μg/mL dealing with 72 h [66]. This result suggested that induction of apoptosis and cell cycle arrest were involved in the pathway. The subset proportion of Th1 cells increased significantly and both Th2 cells and Th17 cells decreased profoundly. In vivo, T lymphocyte infiltration in tumor-burdened mice was enhanced with treatment. Methanol extracts from *Allium atroviolaceum* upregulated caspase-3, -5, and -9 and downregulated Bcl-2, CDK1, and p53 in HeLa cells at concentrations of 20 μg/mL, 40 μg/mL, 60 μg/mL, 80 μg/mL, and 100 μg/mL for 24 h, 48 h, and 72 h [67]. The efficacy was the best at 72 h. Controlling the mechanisms, it inhibited cell growth and proliferation, induced cell cycle arrest, and reduced cell viability. The level of caspase-3 was increased and the level of PARP-1 was decreased by methanol extracts from *Corylus avellane* in HeLa cells at dose of 250 μg/mL and 500 μg/mL for 24 h [68]. The expression of cleaved forms of caspase-3 and PARP-1 suggested that the extracts induced apoptosis through caspase-3 activation in HeLa cells. In conclusion, this result suggested that reduction of cell viability, induction of apoptosis, inhibition of oxidant and proliferation were involved in the pathway. Methanol extracts isolated from *Cyperus rotundus* induced apoptosis and DNA fragmentation and inhibited migration in HeLa cells at doses of 25 μg/mL, 50 μg/mL, and 100 μg/mL for an incubation time of 24 h and 48 h [69]. Cytotoxic effects of methanol extracts on the tested cancer cell lines ranged from 4.52 ± 0.57 μg/mL^−1^ to 9.85 ± 0.68 μg/mL^−1^. Moreover, methanol extracts showed anticancer and antimigration activity and it also induced nuclear fragmentation in HeLa cells. Expression of Bax, BAD, caspase-3, p21, and p53 were increased and Bcl-2 was decreased in HeLa cells by treatment of methanol extracts isolated from *Polyalthia longifolia* at dose of 22 μg/mL for 6 h, 12 h, 24 h, 36 h [70]. Controlling the mechanisms, it altered cell morphology and induced apoptosis. Consequently, extracts produced distinctive porphological features of HeLa cell death that corresponds to apoptosis. Sul’ain et al. reported that the methanol extracts of *Pyrrosia piloselloides* showed antiproliferative effects on HeLa cells [71]. This efficacy was caused by treatment with an IC_50_ of 16.25 μg/mL. Meanwhile, *Pyrrosia piloselloides* water extracts were without influence. Panicker et al. reported that methanol extracts from *Teucrium mascatense* were shown to activate caspases and PARP on HeLa cells, following treatment with 25 µg/mL, 50 µg/mL, 125 µg/mL, and 250 µg/mL, for 72 h [72]. In addition, cell rounding, shrinkage, and detachment from other cells were shown by methanol extracts. This result suggested apoptosis and alteration of cell morphology were related to the pathway.

Most studies on apoptosis of extracts were confirmed by mechanisms such as caspase, Bcl-2, Bax, and PARP. Of the 19 extracts that induce apoptosis in cervical cancer, the expression of caspase was most common in 10, but c-caspase expression was higher than that of the compounds. The expression of c-caspase-3 was found in five extracts, c-caspase-8 and caspase-3 in four, caspase-9 in two, and c-caspase-7, -9, and caspase-5 was derived from one each. There were six extracts that inhibit Bcl-2, four extracts that increase Bax, and one extracts that decreases it. There were four extracts that regulated the expression of PARP, which played a firm role in DNA repair processes and developed as chemotherapy sensitizers for cancer treatment [73]. There were two studies performed both in vivo [61,66]. Most of the studies have revealed the mechanism of apoptosis, but few studies have found it [56,69,71]. There were also studies concerned about toxicity due to experiments conducted at high concentrations [56,57,61,65,66,68,72].

### 3.2. Anti-angiogenesis

Angiogenesis is a major cause in the development and metastasis of a variety of tumor types [74]. Local angiogenesis provides oxygen and essential nutrients to the growing tumor, supports tumor expansion and invasion into nearby normal tissue, and is essential for distant metastasis [75]. To be specific, in cervical cancer, angiogenesis plays a leading role in initiation, proliferation, and progression, and also relates to blocking p53 and stabilizing hypoxia-inducible factor-1α, which led to expression of VEGF [76]. So, preventing angiogenesis could be significant in treatment of the disease. Four natural products have this ability toward HeLa cells and CaSki cells (Table 3). When HeLa cells were treated by *Praecitrullus fistulosus* lectin protein (PfLP), which was grown and consumed in subtropical countries, MMP-2 and -9 became downregulated and it led to the induction of apoptosis and inhibition of angiogenesis [77]. This mechanism was efficient at a dose of 50 µg/mL with 24 h. In vivo, anticancer and anti-angiogenic properties of PfLP were observed at dose of 10 mg/kg on day 7, 9, and 11 after the tumor cell transplantation. Moreover, it did not show any side effects or secondary complications in the study. Purified Flaxseed hydrolysate (PFH), extracted from lignan, was shown to induce apoptosis and inhibit angiogenesis and metastasis on HeLa cells [15]. This process was caused by increasing caspase-3 and downregulating MMP-2 and VEGF. The efficient dose was 17.4 µg/mL, dealing with a timeframe of 48 h. These results suggest that PFH could be great treatment with its anticancer activity. Based on our findings, Kuriakose et al. reported that expression of Bax, p53, and Apaf-1 were increased and Bcl-2 was decreased in HeLa cells by treatment of ethyl acetate extracts of *Penicillium sclerotiorum* from *Cassia fistula* L. [63]. Hexadecanoic acid, oleic acid, and benzoic acid were the major active parts in this treatment. Through controlling mechanisms, it could induce cell cycle arrest and inhibit angiogenesis at a dose of 7.75 µg/mL. Seifaddinipour et al. reported that ethyl acetate extracts isolated from *Pistacia vera* L. downregulated TNF, Bcl-2, IAP, and TRAF in CaSki cells [78]. Through the mechanisms, it induced apoptosis and inhibited angiogenesis. The efficient dose was 81.17 ± 2.87 µg/mL, dealing with a timeframe of 72 h.

A total of four natural products exhibited anti-angiogenesis, and the mechanisms were very diverse. MMP performs a complex and important role in cancer growth and metastasis [79], and two substances inhibit it. Bcl-2 was inhibited by two substances, and in addition, it showed an anti-angiogenesis effect through a mechanism that inhibits factors such as VEGF, TNF, and IAP. In the study using *Penicillium sclerotiorum*, the concentration of the substance (7.75 µg/mL) was significantly lower than that of other anti-angiogenesis studies [63]. By increasing the expression of Bax, p53, and Apaf-1 and inhibiting Bcl-2, it was shown to induce not only anti-angiogenesis, but also cell cycle arrest and apoptosis at doses of 5 µg/mL, 25 µg/mL, and 50 µg/mL.

### 3.3. Anti-Metastasis

Metastasis is the propagation of transformed cells from the organ of origin to other parts of the body and the successive proliferation of tumor colonies [80]. The mechanism includes complicated processes, such as cancer cell detachment from extracellular matrix, migration, invasion, and extravasation to the circulation, and most cancer patients die from metastasis rather than primary tumors [81]. Six natural products including EGCG inhibited metastasis (Table 4). The chemical structures of compounds are shown in Figure 2.

Expression of E-cadherin was increased and p38 and PI3K were decreased in SiHa cells by treatment of astragaloside IV from Radix Astragali [82]. Additionally, astragaloside IV demonstrated inhibition of cell metastasis on SiHa cell lines. The efficient dose was 200 µg/mL, dealing with a timeframe of 24 h. Expression of MMP-2, -9, and VEGF was decreased in HeLa cells by treatment of epigallocatechingallate (EGCG) from green tea [83]. Thus, it induced apoptosis at a dose of 50 µg/mL for 48 h. In addition, it inhibited cell metastasis and proliferation in cervical cancer cells by reducing VEGF, CDK2, and ERK1/2. Praeruptorin-B isolated from *Peucedanum praeruptorum* Dunn. downregulated NF-κB, MMP-2, and -9 in HeLa and SiHa cells [84]. Inhibition of cell metastasis could be observed following the treatment of praeruptorin-B at doses of 40 µg and 60 µg over 24 h. Moreover, it blocked Akt phosphorylation without affecting the MAPK pathway. These results suggested that praeruptorin-B could be good at anticancer activity, especially in cervical cancer cells. Thymoquinone from *Nigella sativa* was shown to induce apoptosis, migration, and invasion on CaSki and HeLa cells [46]. It was efficient when CaSki cells and HeLa cells were treated at a concentration of 5 µM for 24 h. Furthermore, it included the mechanism of increasing E-cadherin level and decreasing the level of Twist1 and Zeb1. Ethanol extracts from *Bauhinia variegata candida* decreased the level of MMP-2 and -9 on HeLa cells [58]. This mechanism was caused when HeLa cells were treated at a concentration of 25 µg/mL for 24 h. Subsequently, it reduced cell viability, migration, and invasion. Lee et al. reported that the decline of MMP-9 and ERK1/2 was observed after the exposure of *Terminalia catappa* ethanol extracts on HeLa and SiHa cells [85]. This process was efficient when the dose was 25 µg/mL, 50 µg/mL, and 75 µg/mL, dealing with a timeframe of 24 h and it led to inhibition of cell metastasis. In conclusion, ethanol extracts blocked the MMP-9 through ERK1/2 pathway by anti-metastatic effects.

A total of six substances derived from natural products inhibited the metastasis of cervical cancer, and MMP control was the most important mechanism. Four substances regulated MMP, MMP-2 was inhibited in three substances, and MMP-9 was inhibited in four substances. E-cadherin is a protein remarkably related to tumor invasion, metastatic transmission, and poor patient prognosis [86]. It was derived from two substances, and in addition, it showed anti-metastasis action through inhibitory mechanisms such as p38, VEGF, and Twist1. In particular, ethanol extracts from *Bauhinia variegate candida* exhibited multi-effect, inhibited migration and invasion at 25 µg/mL, and also induced apoptosis at 15 µg/mL [58]. However, the concentration of the natural product was so high (200 µg/mL) that there was a study involving cytotoxicity concerns [82].

### 3.4. Drug Resistance

Drug resistance is multifaceted, which is due to tumor heterogeneity, tumor microenvironment, and so forth [87]. It can exist against any effective anticancer drug and can develop in several mechanisms, and unfortunately, in a respectable number of patients, tumors do not respond to the treatment [88]. 5 studies reported the natural products that can regulated the drug resistance (Table 5 and Figure 3), and chemical structures of compounds sensitizing drug resistance are shown in Figure 4.

When HeLa cells were treated by rosin abietane diterpenoid, which was isolated from pine rosin, the level of caspase-3 and Bax were increased but Bcl-2 was decreased [89]. Specifically, the level of Th1 cells were increased and both TH2 cells and TH17 cells were decreased. These processes were efficient when HeLa cells were treated at concentration of 1.08 ± 0.12 µM. Finally, it induced apoptosis and cell cycle arrest in HeLa cells. Levrier et al. suggested that thalicthuberine from *Hernandia albiflora* downregulated tubulin polymers on HeLa cells [90]. They used HeLa cells expressing the end-binding protein (EB1-GFP) with thalicthuberine. Through this mechanism, it induced apoptosis and cell cycle arrest at a dose of 2.5 µM for 72 h. Thymoquinone from *Nigella sativa* was shown to have apoptotic effect and inhibition of proliferation on SiHa and CaSki cells [46]. This efficacy was made by upregulating E-cadherin level and downregulating the level of Twist1 and Zeb1; a dose of 5 µM and 10 µM for 24 and 36 h showed the best efficacy. Levels of caspase-3 and -8 were increased on HeLa cells by treatment of 6α-acetoxyanopterine (6-AA) from *Anopterus macleayanus* [91]. It induced cell apoptosis and inhibition of proliferation through mitosis, severe mitotic spindle defects, asymmetric cell divisions, and finally mitotic catastrophe. These processes mostly occurred at concentrations of 3.2 nM and 11.6 nM for 48 h. Huang et al. reported that lipid-soluble extracts from *Pinellia pedatisecta* Schott. up-regulated P53 and downregulated ERK, Bcl-2, PCNA, and HPV E6 [66]. It yielded induction of apoptosis on CaSki and HeLa cells, at a dose of 5000 µg/mL for three days.

We reviewed a total of five studies and the most common factors affecting multi-drug resistance (MDR) were expression of caspase and inhibition of Bcl-2. Caspase expression was increased in two natural substances, both substances increased the expression of caspase-3, and caspase-8 was induced from one substance. The regulation of MDR through Bcl-2 inhibition appeared in two substances, and expressions of p53, Bax, and E-cadherin were also found in one substance each. Especially, thymoquinone obtained from *Nigella sativa* affects apoptosis, anti-metastasis, and MDR [46]. At a low concentration of 5 µM, it showed apoptosis induction and inhibitory effects of migration and invasion by increasing E-cadherin and decreasing Twist1 and Zeb1, at 5 µM and 10 µM, apoptosis, proliferation, and MDR were controlled through the same mechanism. At concentrations of 10 µM, 20 µM, and 40 µM, the effects of apoptosis, anti-migration, anti-invasion, and anti-growth were found, by increasing Bax and E-cadherin. Additionally, the lipid-soluble extracts of *Pinellia pedatisecta* Schott. induced cell cycle arrest and apoptosis at 500 µg/mL, and affects apoptosis and MDR at 5000 µg/mL [66]. However, it should be considered that the experiment is done by an extremely high dose.

### 3.5. MicroRNA Regulation

MicroRNAs, short non-coding RNAs that could regulate gene expression, could regulate various cellular processes [92]. They perform in a feedback structure by protecting key biological processes including cell proliferation, differentiation, and apoptosis, and regulate tumor formation and the development and progression of malignant tumors [93]. It is possible since miRNAs can target many genes, and also one gene can be targeted by different miRNAs [94]. There are important scientific studies showing the usefulness of miRNAs as biomarkers for prediction, diagnosis, and prognosis, and there is also evidence that suppression of oncogenic miRNAs or replacement of tumor suppressor miRNAs could be used to develop new therapeutic strategies [95].

Two studies were reported to suppress the cervical cancer via miRNA regulation (Table 6 and Figure 5). 1′S-1′-acetoxychavicol acetate (ACA) from *Alpinia conchigera* upregulated SMAD4 and downregulated miR-210, which led induction of apoptosis and inhibition of proliferation on SiHa and CaSki cells [20]. It was evident at doses of 20 µM and 30 µM for 6 h and 12 h. The result of study suggested that combination of miRNAs could be a new strategy in treating cervical cancer. Phuah et al. elucidated that ACA from *Alpinia conchigera* induced apoptosis and inhibited proliferation by increasing caspase-3 and -8 and decreasing miR-629 in SiHa and CaSki cells [48]. It was evident at concentrations of 20 µM and 30 µM for 6 h, 12 h, and 48 h. This result suggested the potential of ACA with miR-629 as a cervical cancer treatment. Furthermore, apoptosis was also observed via increasing the expression of RSU1 and GAPDH at the same concentrations.

## 4. Previous Studies

Cervical cancer is the fourth leading cause of cancer death in women, the most commonly diagnosed cancer in 28 countries, and the leading cause of cancer death in 42 countries [2]. Surgery, radiotherapy, and chemotherapy are being performed for cervix cancer, but side effects and complications can occur, so a new potent treatment for cervical cancer by natural products is attracting attention. In 2002, de MOURA et al. reported a review article about natural products that inhibit cervical cancer, however it was a decade ago [96]. A systematic review paper was published in 2013 that analyzed 20 clinical trials; retinoids, vitamin E, probiotics, indoles, multivitamin, folic acid, and selenium were summarized, but the number of substances was not large, and since the paper was published in 2013, the latest research is not included [97]. Wang et al. reviewed plants that have various effects on cervical cancer, divided by their mechanism including apoptosis, cell cycle arrest, anti-migration, and anti-invasion [98]. However, this study was also published in 2013 and does not include anti-angiogenesis, multi-drug resistance, and miRNAs. Lastly, there was a study that introduced natural treatments, which summarized in vitro and in vivo studies on the effects of various substances on cervical cancer and substances that enhance the efficacy of anticancer and radiation therapy [99]. However, this study was published in 2015 and is not organized according to each mechanism. Therefore, it is necessary to investigate the latest research on natural products effective in cervical cancer, with systematic review and mechanism arrangement of the processes such as apoptosis.

## 5. Limitation and Strength

The limitation of our review was that studies on decoctions and that are older than three years are not included, and clinical trials were not searched. Although natural products were effective in inhibiting cervical cancer cells, there are still variations in stabilization and toxicity. In the above, the latest studies of natural products with various effects on cervical cancer were examined, and various mechanisms, concentrations, and experimental models were reviewed. The ultimate goal of our study is to find new therapeutic substances that are effective against cervical cancer and have fewer side effects. Because natural products have fewer side effects, they may have potential as an alternative therapy for cervical cancer. In order to use natural products in clinical practice, additional research is needed to find the most effective and minimally toxic doses for humans. Studies on the synergistic effects of natural products and cervical cancer surgery, radiation, and chemotherapy are also needed. We look forward to further research on finding substances with excellent anticancer effects and hope that this research will be the basis for finding new treatment substances for cervical cancer.

## 6. Conclusions

Our study reviewed the natural products that have anti-tumor effects on cervical cancer. Through various mechanisms, 64 natural products exhibited apoptosis, anti-angiogenesis, anti-metastasis effects and regulated multi-drug resistance and miRNAs. Most of the natural products (47) induced apoptosis including curcumin, emodin, and *Penicillium sclerotiorum*. Four natural products, including Lignan and *Pistacia vera* L., showed anti-angiogenic property in cervical cancer cells. EGCG and five other natural products inhibited metastasis of cervical cancer. Five studies reported that pine rosin and other natural products sensitized drug resistance in cervical cancer. There are few studies that demonstrated miRNA regulating effect of natural products. We expect the non-clinical findings of this study to serve as the groundwork for finding novel cervical cancer treatments that can be used in clinic with less side effects.

## Figures and Tables

**Figure 1 nutrients-13-00154-f001:**
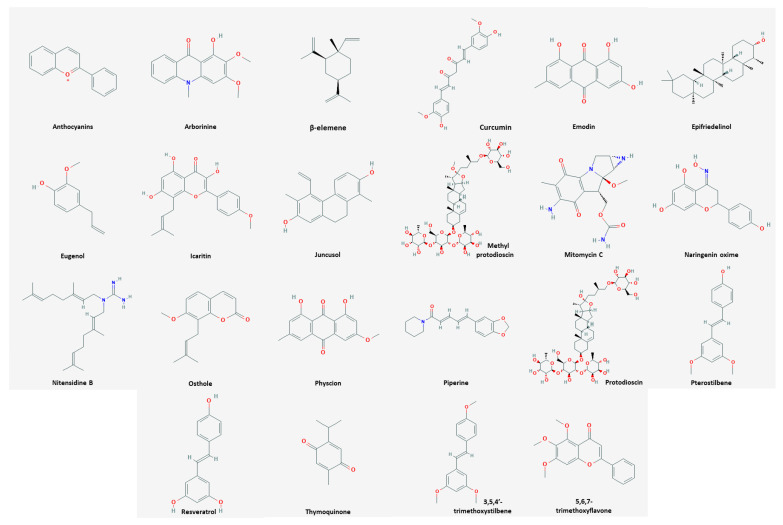
Chemical structures of compounds derived from natural products inducing apoptosis.

**Figure 2 nutrients-13-00154-f002:**
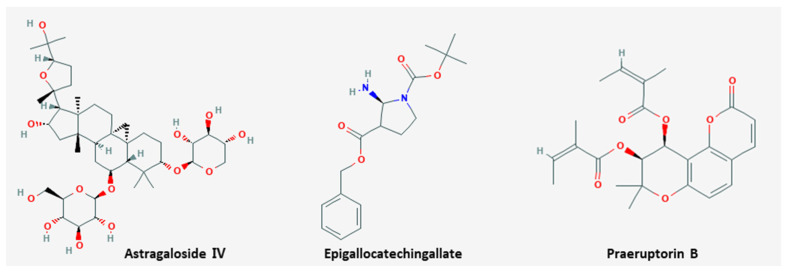
Chemical structures of compounds derived from natural products inhibiting metastasis.

**Figure 3 nutrients-13-00154-f003:**
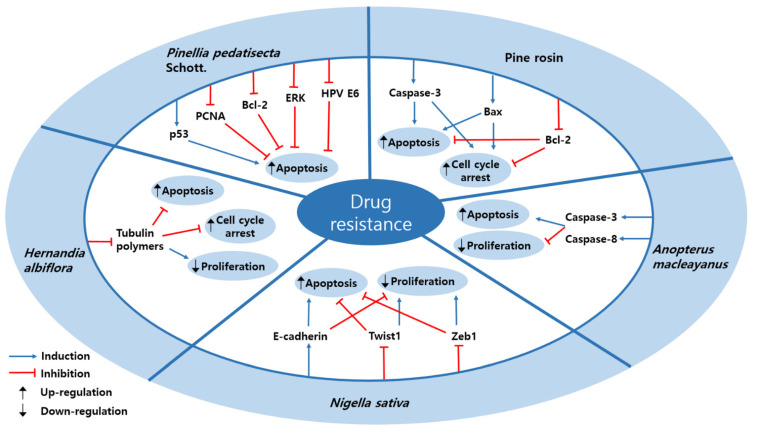
Schematic diagram of drugs resistance signal pathways regulated by natural products in cervical cancer.

**Figure 4 nutrients-13-00154-f004:**
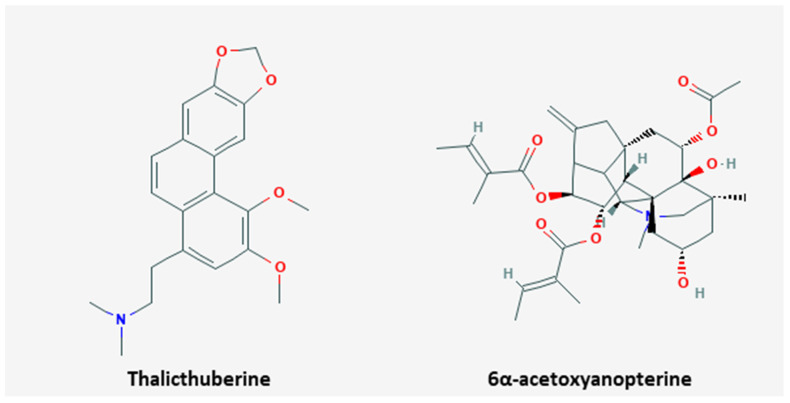
Chemical structures of compounds derived from natural products sensitizing drug resistance.

**Figure 5 nutrients-13-00154-f005:**
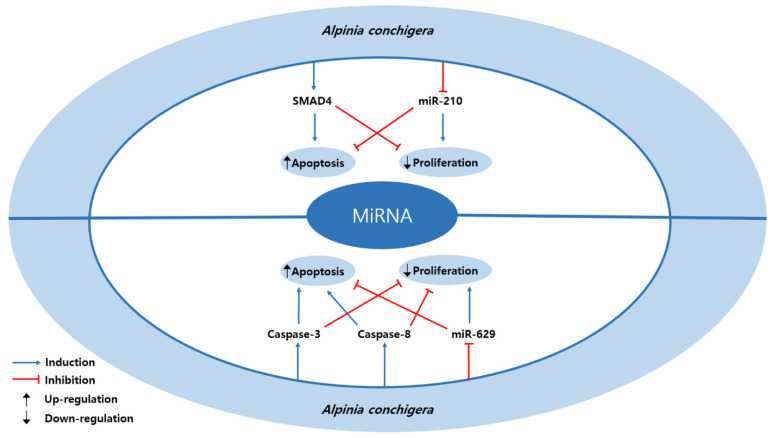
Schematic diagram of miRNAs regulated by natural products in cervical cancer.

**Table 1 nutrients-13-00154-t001:** Apoptosis inducing natural products-compounds.

Classification	Compound	Source	Cell Line/ Animal Model	Dose; Duration	Efficacy	Mechanism	Reference
Etc.	Acylhydrazone		HeLa	2.21 µM; 48 h	Inhibition of cancer activity		[23]
Plant	Anthocyanins	Root tubers and leaves of *Ipomoea batatas*	HeLa	100, 200 µg/mL; 48 h	Induction of apoptosis, cell cycle arrest	↑CFP/YFP	[24]
Plant	Arborinine	*Glycosmis parva*	HeLa	110 μg/mL; 24 h	Induction of apoptosis Inhibition of migration	↑caspase-3, -7 ↓Bcl2-L1	[25]
Plant	β-elemene	*Curcuma zedoaria*	SiHa	30, 40, 50 μg/mL; 24, 48, 72 h	Inhibition of proliferation and migration Induction of cell cycle arrest and apoptosis	↑p15, p53, Bax ↓cyclin D1, Bcl-2, MMP-2, -9, β-catenin, TCF7, c-Myc	[26]
Plant	Copper oxide nanoparticles	*Azadirachta indica, Hibiscus rosa-sinensis, Murraya koenigii, Moringa oleifera, Tamarindus indica*	HeLa	2, 5, 10, 25, 50, 100 μg/mL; 48 h	Inhibition of oxidative stress Induction of apoptosis		[27]
Plant	Curcumin	*Curcuma longa*	HeLa, C57BL/6, BALB/c	In vitro: 2 μg/mL; 48 h In vivo: 25 mg/kg	Induction of apoptosis and cell cycle arrest	↑p53, cytochrome c, PARP, caspase-3, -7, -9 ↓Bcl-2, NF-κB	[28]
Plant	Emodin	*Rhamnus sphaerosperma* var. *pubescens*	SiHa, C33A	46.3, 92.8, 185 μg/mL; 6, 12, 24 h	Induction of, apoptosis	↓NO-, O_2_-, HOCl/OCl-, p-Akt	[29]
Plant	Epifriedelinol	*Aster tataricus*, *Vitex peduncularis* Wall.	HeLa	50, 100, 250, 500, 1000 μg/mL; 72 h	Induction of apoptosis	↑caspase-3, -8, -9 ↓Bcl-2, -xL, survivin	[30]
Plant	Eugenol	*Syzygium aromaticum*	HeLa, SiHa	12.5, 25 µM; 24, 48 h	Induction of apoptosis	↑Bax, PARP, caspase-3, ROS ↓Bcl-2, XIAP	[31]
Plant	Icaritin	*Epimedium*	HeLa, SiHa	HeLa: 12.5, 25 µM; 24, 48, 72 h SiHa: 17, 34 µM; 24, 48, 72 h	Induction of apoptosis Inhibition of proliferation	↑ROS, Bax, c-caspase-3, -9 ↓Bcl-2, XIAP	[32]
Plant	Juncusol	*Juncus inflexu*	HeLa, SiHa, CaSki	1, 3, 10, 30 µM; 24, 48, 72 h	Induction of apoptosis Inhibition of proliferation	↑caspase-3, -8, -9 ↓EGFR, tubulin polymerization	[33]
Plant	Methyl protodioscin	Rhizoma of *Polygonatum sibiricum*	HeLa	18.31, 40, 49 µM; 24 h	Induction of apoptosis and cell cycle arrest Inhibition of proliferation	↑ ROS	[34]
Plant	Mitomycin C	Ginger, Frankincense	HeLa	10 µg/mL; 24 h	Induction of apoptosis Inhibition of proliferation		[35]
Etc.	Naringenin oxime		HeLa, SiHa	HeLa: 12, 24 µM; 24 h SiHa: 18, 36 µM; 24 h	Induction of apoptosis Inhibition of proliferation	↑caspase-3	[36]
	Naringenin oxime ether						
Plant	Nitensidine B	Leaves of *Pterogyne nitens* Tul.	HPV16, SiHa	30, 60, 120 µM; 6, 12, 24 h	Induction of apoptosis	↑caspase-3, -7 ↓aldolase A, alpha-enolase, pyruvate kinase, glyceraldehyde 3-p-dehydrogenase	[37]
Plant	Notoginsenoside R7	*Panax notoginseng*	HeLa, BALB/c	In vitro: 5, 10, 20, 40 μM; 24, 36, 48 h In vivo: 5, 10 mg/kg	Induction of apoptosis Inhibition of proliferation	↑Bax, p-PTEN, Akt ↓Bcl-2, -xL, caspase-3, -9, raptor	[38]
Plant	Osthole	*Cnidiummonnieri* (L.) Cusson	HeLa, SiHa, C-33A, CaSki	40, 80, 120, 160, 200, 240 µM; 24, 48 h	Induction of apoptosis Inhibition of proliferation	↑Bax, c-caspase-3, -9 proteins, E-cadherin, H2AX ↓Bcl-2, MMP-2, -9, β-catenin, vimentin, N-cadherin, IKKα, p-IKKα, p65, p-p65, p50, NF-κB	[39]
Plant	Physcion	*Rhamnus sphaerosperma* var. *pubescens*	SiHa, C33A	43.8, 87.5, 175 μg/mL; 6, 12, 24 h	Induction of apoptosis	↓HOCl/OCl-, p-Akt	[29]
Plant	Phyto-synthesis of silver nanoparticles	Garlic, Green tea, Turmeric	HeLa	2, 5, 10, 25, 50, 100 μg/mL; 48 h	Induction of apoptosis	↓free radical	[40]
Plant	Piperine	*Piper nigrum* L.	HeLa, PTX	50 µM; 6, 24, 72 h with paclitaxel	Induction of apoptosis	↑Bax, Bcl-2, c-PARP, caspase-3 ↓p-Akt, Mcl-1	[41]
Plant	Prenylflavonoids C1	*Mallotus conspurcatus*	HeLa	30 μM; 24 h	Induction of apoptosis	↑EGFP, ROS, Bcl-2, cytochrome c, Apaf-1, caspase-3, -9 ↓c-Myc, hTERT	[42]
	Prenylflavonoids C5			10 μM; 24 h			
Plant	Protodioscin	*Dioscoreae rhizome*	HeLa, C33A	4 μM; 24, 48 h	Induction of apoptosis and mitochondrial dysfunction	↑JNK, p38, PERK, ATF4, Bax, caspase-3, -8, -9, PARP ↓Bcl-2	[43]
Etc.	Pterostilbene		HPV E6, TC1, C57Bl/6	In vitro: 30 µM; 48 h In vivo: 1 mM; 5 days	Induction of cell cycle arrest	↑caspase-3 ↓PCNA, VEGF	[44]
	Resveratrol						
Plant	Tf-CT-ME	*Tripterygium wilfordii*	HeLa	0.5, 1, 2 µg/mL; 24 h	Induction of cell cycle arrest and apoptosis Inhibition of proliferation	↑c-caspase-3 ↓Bcl-2/Bax	[45]
Seed	Thymoquinone	*Nigella sativa*	SiHa, CaSki	10, 20, 40 μM; 24, 36, 48 h	Inhibition of migration and invasion	↑Bax, E-cadherin ↓Bcl-2, Twist1, vimentin	[46]
Plant	Triphala	*Terminalia chebula* Retz., *Terminalia bellerica* (Gaertn) Roxb., *Phyllanthus emblica* Linn.	HeLa	25-150 μg/mL; 48 h	Induction of apoptosis	↑ERK, p53 ↓c-Myc, cyclin D1, p-Akt, p-NF-κB, p56, p-p44/42, MAPK	[47]
Plant	1′S-1′-acetoxychavicol acetate	*Alpinia conchigera*	CaSki, SiHa	20, 30 μM; 6, 12, 48 h	Induction of apoptosis	↑RSU1, GAPDH	[48]
Plant	2D of oleanolic acid and glycyrrhetinic acid	*Ligustri Lucidi Fructus, Glycyrrhiza uralensis*	HeLa	2, 4 µM; 24, 48 h	Induction of apoptosis Inhibition of proliferation	↑ROS	[49]
	3O of oleanolic acid and glycyrrhetinic acid			1, 2 μM; 48 h			
Etc.	3,5,4′-trimethoxystilbene		HeLa	10 µM; 48 h	Induction of apoptosis		[50]
	5,6,7-trimethoxyflavone						
Plant	5′-*epi*-SPA-6952A	*Streptomyces diastatochromogenes*	HeLa	2, 4, 8, 16 µg/mL; 24 h	Induction of apoptosis and cell cycle arrest Inhibition of proliferation	↑Bax/Bcl-2, cytochrome c, caspase-3, -9, c-PARP, p53 ↓MMP	[51]

Cyan fluorescent protein (CFP); yellow fluorescent protein (YFP); Bcl-2-like1 (Bcl2-L1); Bcl-2 associated X protein (Bax); B-cell lymphoma 2 (Bcl-2); matrix metalloproteinase (MMP); transcription factor (TCF); cellular myelocytomatosis oncogene (c-Myc); cytochrome complex (cytochrome c); poly (ADP-ribose) polymerase (PARP); nuclear factor kappa-light-chain-enhancer of activated B cells (NF-κB); phospho-Akt (p-Akt); B-cell lymphoma-extra large (Bcl-xL); reactive oxygen species (ROS); x-linked inhibitor of apoptosis protein (XIAP); cleaved caspase (c-caspase); epidermal growth factor receptor (EGFR); 3-phosphate-dehydrogenase (3-p-dehydrogenase); phospho-phosphatase and tensin homolog (p-PTEN); epithelial cadherin (E-cadherin); H2A histone family member X (H2AX); neural cadherin (N-cadherin); inhibitor of NF-κB kinase α (IKKα); phospho- IKKα (p-IKKα); phospho-p65 (p-p65); cleaved PARP (c-PARP); myeloid cell leukemia sequence 1 (Mcl-1); enhanced green fluorescent protein (EGFP); apoptotic protease activating factor 1 (Apaf-1); human telomerase reverse transcriptase (hTERT); c-Jun N-terminal kinases (JNK); protein kinase RNA-like endoplasmic reticulum kinase (PERK); activating transcription factor 4 (ATF4); proliferating cell nuclear antigen (PCNA); vascular endothelial growth factor (VEGF); transferrin-modified microemulsion carrying coix seed oil and tripterine (Tf-CT-ME); extracellular signal-regulated kinases (ERK); phospho- NF-κB (p-NF-κB); phospho-p44 (p-p44); mitogen-activated protein kinase (MAPK); Ras suppressor protein 1 (RSU1); glyceraldehyde 3-phosphate dehydrogenase (GAPDH).

**Table 2 nutrients-13-00154-t002:** Apoptosis inducing natural products-extracts.

Classification	Extract	Source	Cell Line/ Animal Model	Dose; Duration	Efficacy	Mechanism	Reference
Plant	Aqueous extract	*Anemone nemorosa*	HeLa	20.33 ± 2.480 μg/mL; 24, 48 h	Induction of apoptosis Inhibition of proliferation	↑PS translocation, c-caspase-3, -8, ROS (24 h) ↓MMP, ROS (48 h)	[55]
Plant	Aril extract	*Strelitzia nicolai*	HeLa	250 μg/mL; 24, 48, 72 h	Inhibition of oxidative stress Induction of apoptosis		[56]
Plant	Ethanol extract	*Astragalus membranaceus*, *Angelica gigas*, *Trichosanthes kirilowii* Maximowicz.	HeLa	100, 200, 400 μg/mL; 24, 48 h	Induction of apoptosis and cell cycle arrest Inhibition of cell viability	↑c-caspase-3, -8, PARP-1 ↓Bax, cyclin D, CDK2, CDK4, CDK6, p27	[57]
Plant	Ethanol extract	*Bauhinia variegate candida*	HeLa	15 μg/mL; 24 h	Inhibition of cell viability and migration Induction of apoptosis	↑c-caspase-3, -8, RIP, TNF-R1 ↓MMP-2, MMP-9	[58]
Plant	Ethanol extract	*Botryidiopsidaceae* species	HeLa	6.25, 12.5, 25, 50 μg/mL; 24 h	Inhibition of oxidative stress and migration Induction of apoptosis	↑p53, c-caspase-3 ↓Bcl-2	[59]
Plant	Ethanol extract	*Chloromonas* species	HeLa	12.5, 25 μg/mL; 24, 72 h	Inhibition of oxidative stress Induction of apoptosis	↑c-caspase-3, p53 ↓Bcl-2	[60]
Plant	Ethanol extract	*Dendrobium chrysanthum*	HeLa, Swiss albino mice	In vitro: 450 μg/mL; 24 h In vivo: 50, 100 mg/kg	Induction of apoptosis	↑Bax, p53 ↓Bcl-2	[61]
Plant	Ethanol extract	*Rhamnus sphaerosperma* var. *pubescens*	SiHa, C33A	25, 50, 100 μg/mL; 6, 12, 24 h	Induction of apoptosis	↓HOCl/OCl-, p-Akt	[29]
Plant	Ethyl acetate extract	*Gynura formosana* Kitam.	HeLa	30 μg/mL; 72 h	Inhibition of proliferation	↑ LC3-II/LC3-I, ↓P62/GAPDH, MCM7/GAPDH	[62]
Fungus	Ethyl acetate extract	*Penicillium sclerotiorum*	HeLa	5, 25, 50 μg/mL; 24 h	Induction of apoptosis and cell cycle arrest	↑Bax, p53, Apaf-1 ↓Bcl-2	[63]
Plant	Ethyl acetate extract	*Streptomyces* species	SiHa	20, 40, 60 μg/mL; 24 h	Induction of apoptosis and autophagy	↑caspase-3, -9, Bax, LC3-Ⅱ ↓PARP, LC3-Ⅰ, Beclin1, p62	[64]
Plant	Extract	Blueberry	SiHa	50 mg/mL; 24 h with 4 Gy radiotherapy	Enhancement of radiotherapy	↑p53 ↓cyclin D, E, p21, survivin	[65]
Plant	Lipid-soluble extract	*Pinellia pedatisecta* Schott.	HPV^+^TC-1, C57BL/6	In vitro: 500 μg/mL; 72, 120 h In vivo: 10, 20 mg/kg	Induction of cell cycle arrest and apoptosis	↑ β-catenin, c-Myc, cyclin D1, PPAR1 ↓Th2, Th17	[66]
Plant	Methanol extract	*Allium atroviolaceum*	HeLa	20, 40, 60, 80, 100 μg/mL; 24, 48, 72 h	Induction of cell cycle arrest	↑caspase-3, -5, -9 ↓Bcl-2, CDK1, p53	[67]
Plant	Methanol extract	*Corylus avellane* L.	HeLa	250, 500 μg/mL; 24 h	Inhibition of oxidative stress Induction of apoptosis	↑caspase-3 ↓PARP-1	[68]
Plant	Methanol extract	*Cyperus rotundus*	HeLa	25, 50, 100 μg/mL; 24, 48 h	Induction of apoptosis		[69]
Plant	Methanol extract	*Polyalthia longifolia*	HeLa	22 μg/mL; 6, 12, 24, 36 h	Induction of apoptosis	↑Bax, BAD, caspase-3, p21, p53 ↓Bcl-2	[70]
Plant	Methanol extract	*Pyrrosia piloselloides*	HeLa,	16.25 μg/mL; 24, 48, 72 h	Inhibition of proliferation		[71]
Plant	Methanol extract	*Teucrium mascatense*	HeLa	25, 50, 125, 250 μg/mL; 72 h	Induction of apoptosis Inhibition of proliferation	↑c-caspase-7, -8, -9, PARP	[72]

Phosphatidylserine (PS); cyclin-dependant kinase (CDK); receptor-interacting protein (RIP); tumor necrosis factor receptor 1 (TNF-R1); light chain (LC); minichromosome maintenance protein complex (MCM); peroxisome proliferator-activated receptor 1 (PPAR1); T helper cell (Th); Bcl-2-associated death promoter (BAD).

**Table 3 nutrients-13-00154-t003:** Angiogenesis inhibiting natural products.

Classification	Compound/ Extract	Source	Cell Line/ Animal Model	Dose; Duration	Efficacy	Mechanism	Reference
Fruit	PfLP	*Praecitrullus fistulosus*	HeLa Swiss Albino mice	In vitro: 50 µg/mL; 24 h In vivo: 10 mg/kg	Induction of apoptosis Inhibition of angiogenesis	↓ MMP-2, -9	[77]
Plant	Purified flaxseed hydrolysate	Lignan	HeLa	17.4 µg/mL; 48 h	Induction of apoptosis Inhibition of angiogenesis and metastasis	↑ caspase-3 ↓ MMP-2, VEGF	[15]
Fungus	Ethyl acetate extract	*Penicillium sclerotiorum*	HeLa	7.75 µg/mL; 24 h	Induction of cell cycle arrest and apoptosis Inhibition of angiogenesis	↑ Bax, p53, Apaf-1 ↓ Bcl-2	[63]
Plant	Ethyl acetate extract	*Pistacia vera* L.	CaSki	81.17 ± 2.87 µg/mL; 72 h	Induction of apoptosis Inhibition of angiogenesis	↓ TNF, Bcl-2, IAP, TRAF	[78]

*Praecitrullus fistulosus* lectin protein (PfLP); inhibitor of apoptosis protein (IAP); TNF receptor-associated factor (TRAF).

**Table 4 nutrients-13-00154-t004:** Metastasis inhibiting natural products.

Classification	Compound/ Extract	Source	Cell Line/ Animal Model	Dose; Duration	Efficacy	Mechanism	Reference
Plant	Astragaloside IV	Radix Astragali	SiHa	200 µg/mL; 24 h	Inhibition of cell metastasis	↑ E-cadherin ↓ p38, PI3K	[82]
Plant	Epigallocatechingallate	Green tea	HeLa	50 µg/mL; 48 h	Inhibition of cell metastasis and proliferation Induction of apoptosis	↓ MMP-2, -9, VEGF	[83]
Plant	Praeruptorin B	*Peucedanum praeruptorum* Dunn.	HeLa, SiHa	40, 60 µM; 24 h	Inhibition of cell metastasis	↓ NF-κB, MMP-2, -9	[84]
Seed	Thymoquinone	*Nigella sativa*	CaSki, HeLa	5 µM; 24 h	Induction of apoptosis, migration and invasion	↑ E-cadherin ↓ Twist1, Zeb1	[46]
Plant	Ethanol extract	*Bauhinia variegata candida*	HeLa	25 µg/mL; 24 h	Inhibition of cell viability, migration and invasion	↓ MMP-2, -9	[58]
Plant	Ethanol extract	*Terminalia catappa*	HeLa, SiHa	25, 50, 75 µg/mL; 24 h	Inhibition of cell metastasis	↓ MMP-9, ERK1/2	[85]

Phosphoinositide 3-kinase (PI3K); Zinc finger E-box-binding homeobox 1 (Zeb1).

**Table 5 nutrients-13-00154-t005:** Drug resistance sensitizing natural products.

Classification	Compound/Extract	Source	Cell Line/ Animal Model	Dose; Duration	Efficacy	Mechanism	Reference
Plant	Rosin abietane diterpenoid	Pine rosin	HeLa	1.08 ± 0.12 μM	Induction of apoptosis and cell cycle arrest	↑ caspase-3, Bax ↓ Bcl-2	[89]
Plant	Thalicthuberine	*Hernandia albiflora*	HeLa	2.5 µM; 72 h	Induction of apoptosis and cell cycle arrest Inhibition of proliferation	↓ tubulin polymers	[90]
Seed	Thymoquinone	*Nigella sativa*	SiHa, CaSki	5, 10 µM; 24, 36 h	Induction of apoptosis Inhibition of proliferation	↑ E-cadherin ↓ Twist1, Zeb1	[46]
Plant	6α-acetoxyanopterine	*Anopterus macleayanus*	HeLa	3.2, 11.6 nM; 48 h	Induction of apoptosis Inhibition of proliferation	↑ caspase-3, -8	[91]
Plant	Lipid-soluble extract	*Pinellia pedatisecta* Schott.	CaSki, HeLa	5000 µg/mL; 3 days	Induction of apoptosis	↑ P53 ↓ ERK, Bcl-2, PCNA, HPV E6	[66]

Human papillomavirus E6 (HPV E6).

**Table 6 nutrients-13-00154-t006:** MicroRNA regulating natural products.

Classification	Compound/Extract	Source	Cell Line/ Animal Model	Dose/Duration	Efficacy	Mechanism	Reference
Plant	1′S-1′-acetoxychavicol acetate	*Alpinia conchigera*	SiHa, CaSki	20, 30 µM; 6, 12 h	Induction of apoptosis Inhibition of proliferation	↑ SMAD4 ↓ miR-210	[20]
Plant	1′S-1′-acetoxychavicol acetate	*Alpinia conchigera*	SiHa, CaSki	20, 30 µM; 6, 12, 48 h	Induction of apoptosis Inhibition of proliferation	↑ caspase-3, -8 ↓ miR-629	[48]

Mothers against decapentaplegic homolog 4 (SMAD4).

## Data Availability

The data presented in this study are openly available in [Therapeutic Potential of Natural Products in Treatment of Cervical Cancer: A Review. *Nutrients*
**2021**, *13*, 154.] at [https://doi.org/10.3390/nu13010154], reference number [99].

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
