# Peer review of "Therapeutic Potential of Natural Products in Treatment of Cervical Cancer: A Review"

_nutrients, 2021, doi:10.3390/nu13010154_

Round 1
Reviewer 1 Report
This is an interesting review about the role of dietary natural products for the treatment of cervical cancer.
I have some minor issues that should be changed before publication:
- You should include subchapters for the different compounds in the Chapter: apoptosis, angiogenesis, metastasis, drug resistence and miRNA. Otherwise the review is too confusing.
- Check the numbering of the figures, there is twice figure 1........
- Figure 2 is not informative. Could you at least subdivide the box "natural products" and group the compounds? May be it is better to put Figure 2 as a graphical abstract.........
- A good summary picture would present the most promising candidates for treatment of cervical cancer.
Author Response
We appreciate reviewers and editors for their kind and careful comments for improving the quality of our manuscript and also sincerely hope we address our responses well to the raised comments
I have some minor issues that should be changed before publication:
- You should include subchapters for the different compounds in the Chapter: apoptosis, angiogenesis, metastasis, drug resistence and miRNA. Otherwise the review is too confusing.
(Response): Thank you for your comments. The number of substances we reviewed is too many to make subchapters for all different compounds. So, we divide the apoptosis chapter into two subchapters: 3.1.1 compounds (page 2) and 3.1.2 extracts (page 13) for readers.
- Check the numbering of the figures, there is twice figure 1........
(Response): Thanks. There was a mistake on numbering. Revised.
- Figure 2 is not informative. Could you at least subdivide the box "natural products" and group the compounds? May be it is better to put Figure 2 as a graphical abstract.........
(Response): Thanks. The figure 2 is erased and we used it as a graphical abstract as your suggestion.
- A good summary picture would present the most promising candidates for treatment of cervical cancer.
(Response): We present five substances which are Penicillium sclerotiorum extracts from Cassia fistula L., ethanol extracts from Bauhinia variegate candida, thymoquinone obtained from Nigella sativa, lipid-soluble extracts of Pinellia pedatisecta Schott., and 1'S-1'-acetoxychavicol extracted from Alpinia conchigera with multi-effects in the abstract. Also, chemical structures of the compounds were added.
We sincerely hope we address our responses well to the raised comments and our revised manuscript would be accepted for publication in your journal soon.
Reviewer 2 Report
Park and colleagues designed an interesting review article addressing the role of dietary natural products in cervical cancer treatment, with special emphasis on mechanisms of action. Through the title reading the manuscript sounds extremely interesting but when looking at abstract and whole body of the manuscript, the authors failed in what really aimed to do.
First of all, abstract is not attractive either representative of the whole manuscript. Why only Pubmed and Google scholar were selected as databases for articles search? are articles given from such databases representative? In addition, why only 1's-1'-acetoxychavicol was detailed as effective, given that a lot of bioactive molecules were listed in the whole manuscript as highly effective?
Section 1 and 2, entitled cervical cancer and natural products, respectively, should be restructured, reorganized and mixed into a section "Introduction". In addition, in l. 4, pag. 1 what the authors what to say with "both sexes and all ages"? Is this type of cancer occurring in males?
Last sentence of the section 2, what was the real focus of this study? to address in vitro, in vivo or clinical studies? Please clarify
What was the methodology used for search? what keywords were selected? this markedly determines the amount of data obtained. In addition, through searching in the Web of Science, a total of 177 articles were published on such matter, of them 153 were original articles and 21 review articles. So, the authors should perform a more detailed and careful search of data published over the last 3 years, and to properly include the methodology used for search.
Section 3, including subsection 3.1., 3.2., 3.3., 3.4., and 3.5. should be completely restructured. The authors should organized data by study type (in vitro vs in vivo), should also consider the different classes of phytochemicals (i.e., phenolic compounds, terpenes, alkaloids, etc.) as well as the type of extracts used, to what specifically concerns to extraction solvents used, as they highly impact the extraction yield and consequently phytochemicals bioactivity. In addition, the authors should highlight the most effective molecules, and to mention the plant families they belong, as most phytochemicals isolated from the same plant families exert similar biological effects. Also worth of note in the plant part from whom the isolated bioactive was isolated.
Tables 1-5 should be completely reorganized. I suggest to organize table by study type, then by phytochemicals class, in alphabetic order, also including the plant part and family from whom was extracted and then by type of biological effect stated
in the whole discussion section no mention was done to phytochemicals structure and its consequent biological activity.
Figures 1 and 2 intends to focus on the mechanisms of action of natural products in cervical cancer. Worth of note is that we cannot address to a plant a specific mechanism of action. A mechanism of action can only be stated to a single bioactive molecule. So I suggest to change figures and to include the biologically active phytochemicals isolated from such plants responsible for the observed effects
Section 4 "Previous studies" this section needs to be completely restructured. Most data is repeated from that presented in the introduction section. In addition, why no mention was done to clinical studies? It is undoubtedly more interesting and important to focus on clinical trials than to merely describe data obtained from pre-clinical studies, given that most biologically active compounds did not reveal the same effects neither effectiveness in humans.
Section 5 do not makes sense here, and should be included in the previous sections.
Section 6 "Limitation and strength" markedly underlined the voluntary decision to exclude clinical trials. Why no mention was done neither the real importance of such studies was highlighted? In addition, and given the current focus of this review (only pre-clinical studies) why no mention was done to nano delivery systems formulation for better drug/phytochemicals effectiveness (including their combination) and targeted therapies? Why the authors did not address such extremely important aspect?
Conclusion section should be completely revised and restructured, it is not representative of the data included here.
Author Response
We appreciate reviewers and editors for their kind and careful comments for improving the quality of our manuscript and also sincerely hope we address our responses well to the raised comments
Park and colleagues designed an interesting review article addressing the role of dietary natural products in cervical cancer treatment, with special emphasis on mechanisms of action. Through the title reading the manuscript sounds extremely interesting but when looking at abstract and whole body of the manuscript, the authors failed in what really aimed to do.
(Response): We were trying to reveal the mechanism of action thus, we put the mechanism part in tables and made figures about the mechanisms. However, according to your comments, maybe the efforts were not enough. We changed the title of this review as “Therapeutic Potential of Natural Products in Treatment of Cervical Cancer: A Review”
First of all, abstract is not attractive either representative of the whole manuscript. Why only Pubmed and Google scholar were selected as databases for articles search? are articles given from such databases representative? In addition, why only 1's-1'-acetoxychavicol was detailed as effective, given that a lot of bioactive molecules were listed in the whole manuscript as highly effective?
(Response): Thanks for your comments. There could be limitations about collecting articles by using Pubmed and good scholar. However, sixty-four research recent articles were reviewed. There are review articles which used only Pubmed or google scholar. East Mediterr Health J . 2020 Dec 9;26(12):1565-1569. doi: 10.26719/emhj.20.065., Bipolar Disord . 2020 Dec 22. doi: 10.1111/bdi.13039. Online ahead of print., Cureus . 2020 Nov 16;12(11):e11509. doi: 10.7759/cureus.11509., Int J Sports Phys Ther . 2020 Aug;15(4):624-642. Acta Trop . 2020 Dec 19;105800. doi: 10.1016/j.actatropica.2020.105800. and so forth. Also, we presented five substances which are Penicillium sclerotiorum extracts from Cassia fistula L., ethanol extracts from Bauhinia variegate candida, thymoquinone obtained from Nigella sativa, lipid-soluble extracts of Pinellia pedatisecta Schott., and 1'S-1'-acetoxychavicol extracted from Alpinia conchigera with multi-effects in the abstract.
Section 1 and 2, entitled cervical cancer and natural products, respectively, should be restructured, reorganized and mixed into a section "Introduction". In addition, in l. 4, pag. 1 what the authors what to say with "both sexes and all ages"? Is this type of cancer occurring in males?
(Response): Section 1 and 2 are reorganized as an introduction section. Also, "both sexes and all ages" is unnecessary, so removed.
Last sentence of the section 2, what was the real focus of this study? to address in vitro, in vivo or clinical studies? Please clarify
(Response): The sentence is revised as your suggestion.
What was the methodology used for search? what keywords were selected? this markedly determines the amount of data obtained. In addition, through searching in the Web of Science, a total of 177 articles were published on such matter, of them 153 were original articles and 21 review articles. So, the authors should perform a more detailed and careful search of data published over the last 3 years, and to properly include the methodology used for search.
(Response): We added a methods chapter (page 2), including descriptions of search methods, keywords and selection criteria. It would be better to use web of science and other system, but unfortunately we only used PubMed and Google Scholar for this study.
Section 3, including subsection 3.1., 3.2., 3.3., 3.4., and 3.5. should be completely restructured. The authors should organized data by study type (in vitro vs in vivo), should also consider the different classes of phytochemicals (i.e., phenolic compounds, terpenes, alkaloids, etc.) as well as the type of extracts used, to what specifically concerns to extraction solvents used, as they highly impact the extraction yield and consequently phytochemicals bioactivity. In addition, the authors should highlight the most effective molecules, and to mention the plant families they belong, as most phytochemicals isolated from the same plant families exert similar biological effects. Also worth of note in the plant part from whom the isolated bioactive was isolated.
(Response): We present five natural products with multi-effects in the abstract (page 1). We divide apoptosis chapter into two subchapters: 3.1.1 compounds (page 2) and 3.1.2 extracts (page 13). And some of the missing plant parts were also marked such as Root tubers and leaves of Ipomoea batatas, Rhizoma of Polygonatum sibiricum, and Leaves of Pterogyne nitens Tul.
Tables 1-5 should be completely reorganized. I suggest to organize table by study type, then by phytochemicals class, in alphabetic order, also including the plant part and family from whom was extracted and then by type of biological effect stated
(Response): Thanks. The table of the apoptosis part also separated into compounds and extracts (table 1, page 7; and table 2, page 15). In the rest of the tables, compounds were sorted up and extracts down (table 3, page 18; table 4, page 20; and table 5, page 22). In addition, we added the missing animal model names such as C57BL/6, BALB/c to the table.
in the whole discussion section no mention was done to phytochemicals structure and its consequent biological activity.
(Response): The figures on chemical structures are added as your suggestion (figure 1, page 12; figure 2, page 20; and figure 4, page 24).
Figures 1 and 2 intends to focus on the mechanisms of action of natural products in cervical cancer. Worth of note is that we cannot address to a plant a specific mechanism of action. A mechanism of action can only be stated to a single bioactive molecule. So I suggest to change figures and to include the biologically active phytochemicals isolated from such plants responsible for the observed effects
(Response): I don’t agree with this comment. The mechanism of plant extract could be addressed. For example, Front Oncol . 2020 Oct 8;10:547392. doi: 10.3389/fonc.2020.547392. eCollection 2020., Saudi Pharm J . 2020 Oct;28(10):1155-1165. doi: 10.1016/j.jsps.2020.08.004. Epub 2020 Aug 14., Heliyon . 2020 Oct 5;6(10):e05088. doi: 10.1016/j.heliyon.2020.e05088. eCollection 2020 Oct., Anticancer Res . 2020 Sep;40(9):5097-5106. doi: 10.21873/anticanres.14513., Front Oncol. 2020 Apr 9;10:491. doi: 10.3389/fonc.2020.00491. eCollection 2020., etc. eludated the mechanism of extract. Also, we can’t assume the active phytochemicals from the studies which reported only the mechanism of plant extract without their active phytochemicals.
Section 4 "Previous studies" this section needs to be completely restructured. Most data is repeated from that presented in the introduction section. In addition, why no mention was done to clinical studies? It is undoubtedly more interesting and important to focus on clinical trials than to merely describe data obtained from pre-clinical studies, given that most biologically active compounds did not reveal the same effects neither effectiveness in humans.
(Response): Thank you for your comment. I understand the importance of clinical trials results. So, we are planning to review the clinical trials for another study soon.
Section 5 do not makes sense here, and should be included in the previous sections.
(Response): Section of ‘Multi-functional natural products’ is removed and included in previous sections as your suggestion.
Section 6 "Limitation and strength" markedly underlined the voluntary decision to exclude clinical trials. Why no mention was done neither the real importance of such studies was highlighted? In addition, and given the current focus of this review (only pre-clinical studies) why no mention was done to nano delivery systems formulation for better drug/phytochemicals effectiveness (including their combination) and targeted therapies? Why the authors did not address such extremely important aspect?
(Response): I agree with you about the importance of nano delivery system. There are several studies on nano delivery systems based on our search criteria. There is no separate chapter for nano delivery systems, but can be found in the manuscript and tables. For example, copper oxide nanoparticles, phyto-synthesis of silver nanoparticles, and transferrin-modified microemulsion carrying coix seed oil and tripterine. However, the theme of this review is not nanoparticle so we didn’t focused on that part. Also, our team submitted a review article about nanotechnology with nanophytochemicals in cancer treatment in another journal last week.
Conclusion section should be completely revised and restructured, it is not representative of the data included here.
(Response): Thanks. Conclusion section is revised.
We sincerely hope we address our responses well to the raised comments and our revised manuscript would be accepted for publication in your journal soon.
Round 2
Reviewer 2 Report
Despite some comments were addressed by the authors, some others are still pending and should be addressed.
Author Response
Dear reviewer 2 We sincerely responded to every comment in last revision. I don't know which parts should be addressed for this time. Can you specify your comments? Also, "Are there appropriate and adequate references to related and previous work?" part was 4 star which is now 3 star. However, the references were revised slightly for better citation. We don't know what to do with this comment.